# EXPLAINING CONTRASTIVE MODELS USING EXEMPLARS: EXPLANATION, CONFIDENCE, AND KNOWLEDGE LIMITS

## ABSTRACT

Explainable AI (XAI) provides human users with transparency and interpretability of powerful "black-box" models. Recent work on XAI has focused on explaining specific model responses by identifying key input features using attribution analysis. Another avenue for explaining AI decisions is to leverage exemplars of training data. However, there are limited investigations on using exemplars to establish metrics for confidence and knowledge limits. Recently, contrastive learning has received increased focus in computer vision, natural language, audio, and many other fields. However, there are very few explainability studies that could leverage the learning process to explain the contrastive models. In this paper, we advance post-hoc explainable AI for contrastive models. The main contributions include i) explaining the relation among test and training data samples using pairwise attribution analysis, ii) developing exemplar-based confidence metrics, and iii) establishing measures for the model knowledge limits. In the experimental evaluation, we evaluate the proposed techniques using the OpenAI CLIP model. The evaluation on ImageNet demonstrates that exemplars of training data can provide meaningful explanations for the decision-making of contrastive models. We observe that the proposed exemplar-based confidence score gives a more reliable, dataset-agnostic probability measure compared to the softmax score and temperature scaling. Furthermore, the OOD detection module of our framework shows significant improvement compared to other state-of-the-art methods (6.1% and 9.6% improvement in AUROC and FPR@95TPR, respectively). The three modules together can give a meaningful explanation of the model decisions made by a contrastive model. The proposed techniques extend the body of science of XAI for contrastive models and are expected to impact the explainability of future foundational models.

## 1 INTRODUCTION

The tremendous success of deep learning for cognitive tasks is driving deployment within safety-critical systems and high assurance applications Esteva et al. (2021); Leo et al. (2019). However, the deployment of neural networks within autonomous vehicles or medical diagnosis systems requires the trust of human users Siau & Wang (2018); Yin et al. (2019). Explainable AI (XAI) is positioned to provide this trust by i) *explaining* model decisions, ii) measuring the *confidence* in model responses, and iii) analyzing if the model is operating within its *knowledge limits* Das & Rad (2020). The most popular technique for explaining the response of an AI model involves using feature highlighting Simonyan et al. (2013); Springenberg et al. (2014a); Kapishnikov et al. (2021). Feature highlighting involves identifying features that greatly contribute to a specific model response through attribution analysis such as integrated gradients Sundararajan et al. (2017), GradCAM Selvaraju et al. (2017), and DeepLift Shrikumar et al. (2017). Explanation by exemplars is another way of explaining the model decision Jeyakumar et al. (2020); Bilgin. & Gunestas. (2021); Garima et al. (2020); Lee et al. (2020). These studies focus on evaluating exemplars to see if the model decision is consistent based on the given test data.

Model confidence has mainly been investigated using Platt and Temperature scaling Platt (2000); Guo et al. (2017). These techniques and their extensions scale the softmax output with a temperature

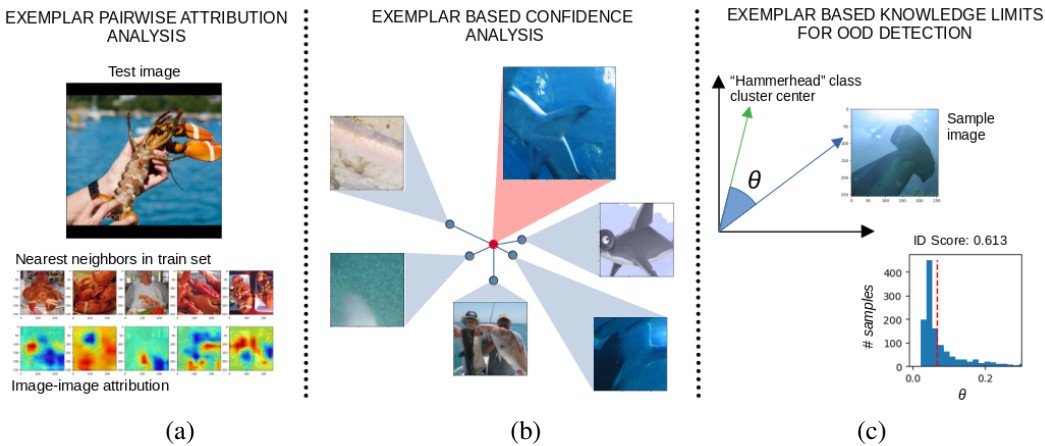

Figure 1: Overview of the exemplar-based Explainable AI (XAI) in the scope of contrastive learning, (a) Linking of test and training data with attributions explaining the linkage. (b) Confidence measure of the model response using exemplar-based distance metrics. (c) Angle-based OOD detection and knowledge limit analysis.

factor that is learned post-hoc or during training Minderer et al. (2021). The knowledge limits of a model can be determined using out-of-distribution (OOD) analysis Yang et al. (2022b); Hsu et al. (2020) which aims to identify if an input is on the manifold of the training data Liang et al. (2020).

The common denominator of these XAI techniques is their computation with respect to an AI model and a specific input. This paper aims to advance post-hoc explainablity for image classifiers through the use of training data exemplars, i.e., explaining responses using both the *model* and examples of *training data*. This is inspired by the success of contrastive learning which associates, or dissociates, exemplars of training data Chen et al. (2020); Radford et al. (2021). Contrastive learning powered the success of modern visual language models (VLMs) Alayrac et al. (2022); Lu et al. (2019); Radford et al. (2021). Existing studies on XAI using exemplars are focused on case-based reasoning. Case-based reasoning identifies examples in the training data that are similar to an input, with the objective of providing an explanation for the model decision Aamodt & Plaza (1994); Rudin et al. (2022). However, state-of-the-art case-based reasoning techniques do not provide effective explainations for why the input and the selected exemplars of training data are similar Salakhutdinov & Hinton (2007); Card et al. (2019). Moreover, exemplar based solutions for measuring the confidence and knowledge limits of AI models have not been investigated.

Our proposed framework answers the three fundamental questions of model explainability: a) "Why does the model think exemplars are relevant to the input?" b) "How confident is the model in its prediction?" and c) "Does the given test data reside in-distribution or is it out-of-distribution for the model scope?". Our main contributions are shown in Figure 1 and can be summarized, as follows:

1. **Exemplar Explanations using Pairwise Attributions:** We explain why pairs of input images are similar using pairwise attribution analysis. The attributions highlight common image features that explain their proximity in the latent space. This can illustrate if the decision made by the model is consistent with the salient regions of the exemplars.

2. **Exemplar-based Confidence:** We propose an exemplar-based confidence measure using $k$ nearest-neighbors distance. The confidence is computed as a weighted sum with respect to the angular distance from exemplars.

3. **Exemplar-based Knowledge Limits:** We pre-characterize the angular distribution of each class. The knowledge limits are next determined using a proposed in-distribution scoring method with respect to the distance from the class centroid.

Together, these modules can show insightful explanations for a given data and the decision provided by a contrastive model. The dashboard-style report focusing on the explainable exemplars, decision confidence, and knowledge limits generated by the proposed framework can be a one-stop solution for the explainability of contrastive models. The remainder of the paper is organized as follows: pre-

liminaries are provided in section 2, the methodology is given in section 3, experimental evaluation with a complete sample report generated by our framework is presented in section 4, and the paper is concluded in section 5.

## 2 RELATED WORK

In this section, we review previous work on contrastive learning, attribution analysis, case-based explanations, confidence metrics, and knowledge limits.

**Contrastive Learning** Contrastive learning is performed by bringing similar pairs of data samples closer together in the latent space and pushing apart dissimilar pairs. Modern contrastive paradigms operate on low-dimensional latent representations Chen et al. (2020); Radford et al. (2021). In SimCLR, data augmentation is performed by constructing multiple data points from a single input sample which are then trained to be similar using contrastive loss Chen et al. (2020). In CLIP, text and image data are aligned using unsupervised contrastive learning Radford et al. (2021). The CLIP model specifically has one image encoder and one text encoder, where the model is trained based on an image-caption dataset. The CLIP framework is the foundation for many subsequent VLMs Alayrac et al. (2022); Lu et al. (2019).

**Attribution Analysis** Attribution methods explain model knowledge by measuring the contribution of input features to model output Simonyan et al. (2013); Ribeiro et al. (2016); Rudin et al. (2022). Due to their speed and quality, backpropagation methods Simonyan et al. (2013); Sundararajan et al. (2017); Springenberg et al. (2014b); Selvaraju et al. (2017) are most commonly used Ancona et al. (2017). Generally, these methods make one forward and backward pass through a network to capture the model gradients for a given input and they can be developed to be agnostic or model-dependent. GradCAM is a popular, model-dependent, backpropagation attribution method typically applied to CNNs Selvaraju et al. (2017). GradCAM computes the gradients of a target class with respect to the last convolutional layer, averages the gradients over the channels, multiplies the averages by the layer activations, and applies a ReLU to retain only those gradients which point to the target class Selvaraju et al. (2017). GradCAM visualizations are shown as a radiating heat map centered on the important features of an input with regard to the classification Selvaraju et al. (2017).

**Case-based Explanations** Case-based explanation is a different avenue of XAI that aims to explain a decision using existing data Aamodt & Plaza (1994). Case-based explanation is a post-hoc method which traditionally finds images that explain black-box model decisions from the training set Rudin et al. (2022). The idea is to find images which share features with a given input to illustrate which features are important Dudani (1976). Modern approaches to case-based explanation find explanations through distance measurement in the latent space. In Deep k-Nearest Neighbors (DkNN), performing kNN with an input and the deep neural network layer representations of training data provides visual explanations for the input Papernot & McDaniel (2018). Other average voting based approaches have also been proposed Card et al. (2019). These latent graphs have for example been used to detect adversarial attacks Papernot & McDaniel (2018); Abusnaina et al. (2021). Case-based explanations are becoming more prevalent with the rise of contrastive learning Chen et al. (2020); Radford et al. (2021).

**Confidence Metrics** Many classification algorithms predict class membership using the probability-like softmax score Goodfellow et al. (2016). The softmax scores have the property of predicting values between 0 to 1 but do not reflect a real probabilistic measure. It is also well known that the scores exhibit poor correlation with prediction confidence Platt (2000); Guo et al. (2017). The scores can however be calibrated into meaningful confidence measures using learnable parameters, which was performed for binary and general classifiers using Platt Platt (2000) and Temperature Guo et al. (2017) scaling respectively. These parameters scale the output logits of all the classes before they are fed to the softmax function. Various approaches have been proposed to learn tunable parameters post-hoc and during training such as averaging predictions Lakshminarayanan et al. (2017); Wen et al. (2020b), augmenting data Thulasidasan et al. (2019); Wen et al. (2020a), and changing from cross-entropy to focal loss Mukhoti et al. (2020).

**Knowledge Limits** Determining the knowledge limits is crucial to ensuring the reliability and safety of machine learning systems. Inputs that are classified to be outside the operating domain can be rejected or handed over to human users for safe processing. Popular approaches to out-of-distribution detection include classification-based, density-based, and distance-based methods Yang et al. (2022b). Classification-based approaches aim to separate in-distribution (ID) and OOD data via modeling softmax score distributions Hendrycks & Gimpel (2016) and further improvements were made by increasing ID and OOD separation through input perturbation Liang et al. (2020). Density-based methods model the probability distribution of ID data such that OOD data can be identified by likelihood Lee et al. (2018). Distance-based metrics detect OOD data by measuring distance from ID data Yang et al. (2022b). This can be done by measuring distance from class centroids via Mahalanobis distance Lee et al. (2018), using non-parametric nearest neighbor distance Sun et al. (2022b), or measuring cosine similarity between ID data and test sample features Techapanurak et al. (2019).

## 3 PROPOSED METHODS

In this section, we provide the details of our proposed techniques. We first show how the exemplars are explained, then how they establish confidence in the response, and finally how they determine knowledge limits.

### 3.1 EXEMPLAR EXPLANATION USING PAIRWISE ATTRIBUTIONS

Exemplars have been used to explain model decisions using case-based reasoning Papernot & McDaniel (2018); Card et al. (2019)Kenny & Keane (2021). However, those studies do not explain why those exemplars are similar, except that the exemplars are closer to the test data in the latent space. One possible solution for explaining similar exemplars is to leverage the concept of prototyping and part prototyping Kim et al. (2014); Li et al. (2018). Prototyping is the concept of finding a small set of representative examples for a class Kim et al. (2014). Part prototyping involves identifying common features between a prototype and a data sample Li et al. (2018).

We extract $k$ exemplar image samples $X_e$ from the training set for a given input image $X_i$. To find the exemplars, the input image and training set are transformed into latent vectors $Z$ via the image encoder $F$ of a contrastive learning model (e.g., CLIP). Then the $k$ nearest neighbors to the input image $X_i$ from the training set in the latent space are selected as the exemplars.

Inspired by part prototyping in Li et al. (2018); Nauta et al. (2020), we are interested in identifying the parts of selected exemplars and the input image that are similar. However, we want to avoid enumerating, or learning, a fixed set of part prototype features ahead of time. Therefore, we propose to leverage attribution analysis to directly identify key features that link an input image and exemplar pair. However, attribution methods are traditionally computed for a single input with respect to the top output logit of a network $F$. In order to leverage standard attribution methods, we define a new output logit from the latent space where exemplars are discovered.

Let $z_i$ be the input sample's latent representation, and $z_e$ be a selected exemplar's latent representation. We define a new output logit $l$ as follows:

$$l = dot(z_e, z_i). \tag{1}$$

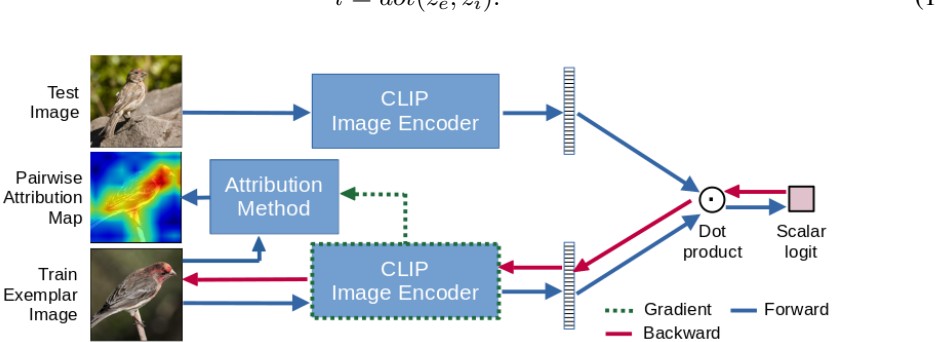

Figure 2: Process of pairwise attribution with a neighbor exemplar image from train-set.

The attribution map $A$, which explains why the pair of samples $(X_e, X_i)$ are similar, can then be defined as follows:

$$A = AttrMethod(F, l), \tag{2}$$

where *AttrMethod* is any attribution method. We now measure the image-image pairwise attribution with respect to the new logit $l$ which represents the images $X_e$ and $X_i$. Note that this formula can be directly applied to any multi-modal model as long as the image encoder is based on convolutions. In our experimental evaluation, we used GradCAM as the attribution method. This pairwise attribution process is illustrated in Figure 2.

## 3.2 EXEMPLAR BASED CONFIDENCE

Confidence scores using Platt and Temperature scaling are based on the relative difference in the output logit for the different classes Platt (2000); Guo et al. (2017). Instead of computing confidence with respect to the raw output logit, we conjecture that it is advantageous to compute confidence as a weighted sum of the inverse distance to the nearest $k$ exemplars. This is partly inspired by non-conformal methods used to detect adversarial examples Papernot & McDaniel (2018). We define the exemplar-based confidence of an input image $X_i$ with respect to class $Y$, as follows:

$$p(Y|X_i) = \frac{\sum_{j \in |K_Y|} 1/\mathcal{L}(z_i, z_e^j)}{\sum_{j \in |K|} 1/\mathcal{L}(z_i, z_e^j)} \tag{3}$$

where $K$ is the set of $k$ nearest neighbors of $X_i$ in the latent space of a contrastive learning model. $K_Y \subseteq K$ is the set of samples in $K$ classified as the class $Y$. $z_i$ is the latent representation of $X_i$. $z_e^j$ is the latent representation of the $j$-th neighbour in $K$ or $K_Y$. $\mathcal{L}$ is the nearest neighbor distance measure.

This equation is defined as a probabilistic measure, such that the weighted average value intrinsically produces a value ranging $[0, 1]$. The weights come from the inverse of the distance between the input test image and the nearest exemplar samples in the latent space. Next, we further extend the proposed confidence technique with a variant of Platt scaling method Platt (2000), as follows:

$$p(Y|X_i) = \frac{\alpha \sum_{j \in |K_Y|} 1/\mathcal{L}(z_i, z_e^j) + \beta}{\sum_{j \in |K|} 1/\mathcal{L}(z_i, z_e^j)} \tag{4}$$

where, $\alpha, \beta \in \mathbb{R}$ are two parameters that can be optimized. These parameters scale and offset the confidence score to better match the accuracy and confidence scores in the reliability analysis. This can contribute to calibrating the probabilistic values to represent the true correctness likelihood. For the uncalibrated exemplar-based evaluations, the values for $\alpha$ and $\beta$ were set to 1 and 0, respectively.

## 3.3 KNOWLEDGE LIMITS

The knowledge limits of a model can be determined using out-of-distribution detection algorithms Hendrycks & Gimpel (2016); Liang et al. (2020); Lee et al. (2018); Sun et al. (2022b); Techapanurak et al. (2019). These algorithms typically utilize the confidence scores (i.e., softmax) or calibrated confidence scores (e.g., Temperature scaling) to detect OOD data samples. Another class of the OOD detection process trains a model using both ID and OOD samples to actively learn the OOD data. Moreover, there are other specialized methods focusing on OOD detection for contrastive models, these are KNN+ Sun et al. (2022a), SSD Sehwag et al. (2021), and CSI Tack et al. (2020).

In this study, we propose an exemplar-based method to detect OOD data. In this process, we predict the class of a given sample within the in-distribution library of classes by estimating the logit vector of the sample with the image encoder. Then, we calculate the angular distance $\theta$ of a logit vector from the mean logits of the predicted class. The angular distance is the inverse cosine of the cosine similarity of the mean logits and the sample logit. Given an input image $X_i$ which is predicted by a contrastive model to be in class $Y$ and produces the logit vector $z_i$, then the angular distance is:

$$\theta = cos^{-1}\left(\frac{z_i \cdot z_Y^C}{||z_i|| ||z_Y^C||}\right) \tag{5}$$

where $z_Y^C$ is the mean logit of all the exemplars in the training set that belongs to class $Y$. Here, $z_Y^C$ is alternatively called the cluster centroid of class $Y$.

Using the angular distance, we pre-characterize the angular distribution for each of the classes in the training data set. We calculate the percentile of the angular distance of the given image $\theta_i$ with respect to the angular distance of the training exemplars $\theta_Y$ belonging to the predicted class of the input image $Y$, as follows:

$$p_i = percentile(\theta_i, \theta_Y). \tag{6}$$

This expression indicates the percent possibility of the sample $X_i$ to be out-of-distribution of the known dataset. The higher the percentile number, the higher the chance of the sample being out-of-distribution.

To calculate the ID score $S_{ID}$ we define the following expression,

$$S_{ID} = 1 - (p_i/100). \tag{7}$$

We then define a threshold $S_{th}$ to decide if the sample resides in or out of distribution. The threshold is calculated by observing the distribution of the known dataset (in-distribution dataset), which can maximize the number of ID samples and the separation of ID and OOD sets. The optimal value of $S_{th}$ for our experiment is reported in section 4.3. Therefore, the binary decision indicating membership in the OOD class for a given image $X_i$ is:

$$OOD(X_i) = S_{ID} < S_{th}. \tag{8}$$

## 4 RESULTS AND EVALUATION

In this section, we present the evaluation of the proposed explainability methodology described in section 3. All experiments for the proposed methods are performed with the ImageNet dataset Russakovsky et al. (2015), implemented with the PyTorch library Paszke et al. (2019), and the experiments are run on NVIDIA A40 GPUs. We use the Captum attribution method library Kokhlikyan et al. (2020) as a reference for the implementation of our own version of image-image pairwise GradCAM attribution Selvaraju et al. (2017). The proposed method is evaluated both quantitatively and qualitatively. We also evaluate the effectiveness of using pairwise GradCAM to explain why the model thought the exemplars were similar to the given input sample.

The contrastively trained OpenAI CLIP Radford et al. (2021) model is used to perform all the experiments in this section. The CLIP model has two encoders - image and text. For reproducibility reasons, we used the pre-trained RN101-based CLIP model from the OpenAI repository. The RN101 model produces a latent vector of length 512. The input size of the image encoder in the CLIP model is 224x224 with three color channels. The input images are preprocessed using the supplied preprocessing function with the RN101 CLIP model, which includes resizing, center-cropping, and normalization. The input of the text encoder model is generated by using the CLIP tokenizer.

### 4.1 EXPLAINING EXEMPLARS USING PAIRWISE ATTRIBUTIONS

We applied our proposed pairwise attribution method to perform the input attribution analysis. Since this method is data type agnostic, we can perform input attribution analysis for any two data points as long as the latent vectors are of the same length. We can estimate neighbor exemplars for a given test image, based on the dot product of the latent vectors representing similarity. To qualitatively examine our input attribution method, we have calculated the input attribution of the neighbor exemplars based on the input test image in Figure 3. This experiment shows the related exemplars in the train set and the relation with the test image (highlighted "Ostriches" in the attribution map). The salient regions in the attribution maps show the most important region for the model to identify the class (Ostrich neck).

### 4.2 EVALUATION OF EXEMPLAR-BASED CONFIDENCE

To evaluate the proposed exemplar based confidence scores described in Section 3.2, we performed reliability analysis adopted from Guo et al. (2017) on the ImageNet, SUN, and aYahoo datasets. The reliability diagrams based on the exemplar and softmax confidence scores are shown in Figure 4. The

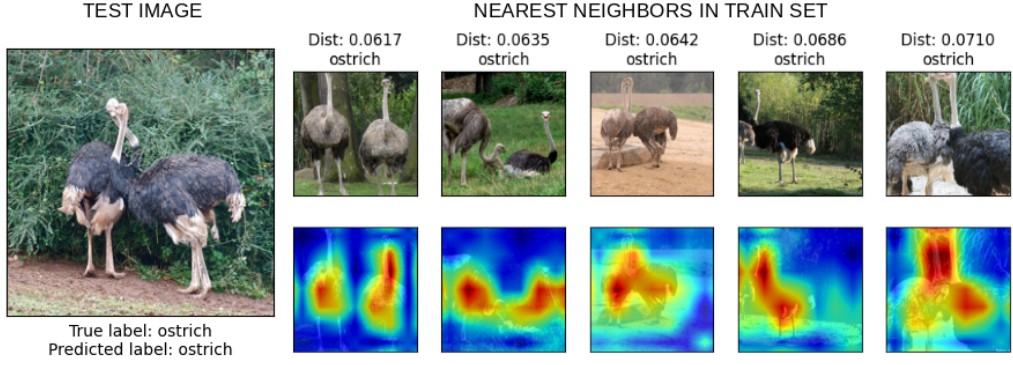

Figure 3: Sample exemplar image-image pairwise attribution with neighbors.

reliability diagram shows the correlation between accuracy and confidence scores in confidence bins. Both the accuracy and confidence values of an ideal classifier should follow the diagonal line of the accuracy-confidence curve. In each confidence bin, the average accuracy and average confidence values of the data points are plotted.

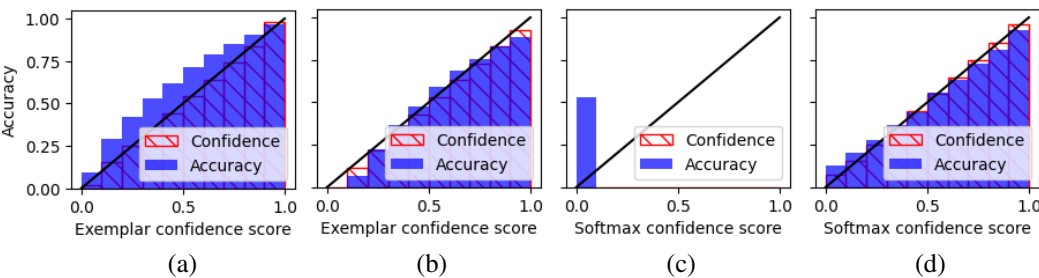

Figure 4: Reliability diagrams for CLIP image encoder model with ImageNet dataset: (a) exemplar-based uncalibrated, (b) exemplar-based calibrated, (c) softmax-based uncalibrated, and (d) softmax-based calibrated scores.

In Figure 4(a), we can see that the uncalibrated exemplar-based confidence score is performing very close to an ideal classifier, rendering mean accuracy values that are aligned with confidence scores in each bin. On the other hand, in fig 4(c), the uncalibrated softmax-based score shows a high number of low-confidence data predictions. This clearly shows that the softmax-based confidence scores by default are not suited for interpreting the latent space of contrastive models. Reliability diagrams for SUN and aYahoo are shown in the appendix section A.1. Those also show similar results as seen on the ImageNet dataset.

To better represent the true probabilistic nature of the confidence scores, we calibrate both the exemplar and softmax-based confidence scores. The softmax score is calibrated using the temperature scaling method Guo et al. (2017). The temperature scaling on the softmax confidence score is calculated by $p(Y|X_i) = softmax(z_i/T)$. Where the $T$ is the temperature parameter. In our experiments for the ImageNet dataset, we have found the optimal values of $\alpha$ and $\beta$ for the exemplar-based confidence score to be 1 and 28.4340, respectively. The optimal $T$ value is found to be 0.0097.

We have also calculated the expected calibration error (ECE) from the reliability diagrams. The expected calibration error measures the disagreement between the accuracy and confidence scores, which is defined by the following equation: $ECE = \sum_i \frac{Bin_i}{N}|a_i - c_i|$. Where, $Bin_i$, $N$, $a_i$, and $c_i$ indicate the number of data points in the $i^th$ bin, the total number of data points, the accuracy of the data points in $Bin_i$, and the average confidence score in $Bin_i$, respectively. Table 1 shows the ECE of the exemplar-based confidence score and the softmax confidence score. Both of these confidence scores are uncalibrated. However, we can see that the exemplar-based confidence score is better than the softmax confidence score as the exemplar-based confidence score resulted in very small ECE values without the need for calibration for all three different datasets. This makes the

proposed exemplar-based confidence metric dataset agnostic, predicting correct probability scores without the need for data calibration. On the other hand, the calibrated exemplar and softmax-based scores resulted in very close ECE values.

Table 1: ECE for exemplar-based confidence score and softmax confidence score with scaling. (Lower is better)

|  | ImageNet | | SUN | | aYahoo | |
|---|---|---|---|---|---|---|
|  | Uncalibrated | Calibrated | Uncalibrated | Calibrated | Uncalibrated | Calibrated |
| Exemplar | **0.085** | 0.029 | **0.070** | 0.035 | **0.022** | **0.020** |
| Softmax | 0.530 | **0.027** | 0.518 | **0.028** | 0.827 | 0.027 |

### 4.3 KNOWLEDGE LIMITS ANALYSIS

To perform the knowledge limits analysis, we first analyzed the ImageNet class distributions using our proposed angular distance metric. Then we introduced the out-of-distribution (OOD) datasets - ImageNet-O Hendrycks et al. (2021), SUN Xiao et al. (2010), and aYahoo Farhadi et al. (2009) to perform the OOD detection using the method described in section 3.3. Near-OOD datasets mimic the data patterns of their original datasets but have a different set of classes, making them out-of-distribution, and making them harder to detect than far-OOD datasets Yang et al. (2022a).

**ImageNet class centroid analysis:** To analyze the class centroid and data points, we illustrate the probability distribution of all data points of a class $Y$ and their angular distances $\theta$ from the mean centroid $z_Y^C$ of that class in Figure 5. We see that the angular distance distributions follow log-normal distribution due to the nature of the cosine similarity-based angular distance calculation.

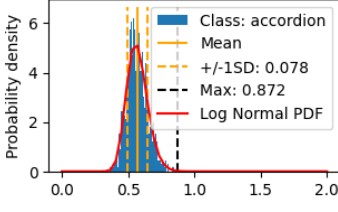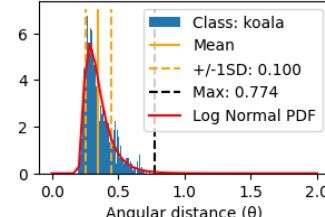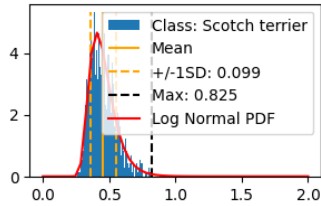

Figure 5: Probability density and fitted log-normal distribution function of classes ("Accordion", "Koala", and "Scotch Terrier") from ImageNet based on angular distance from the class centroid.

**OOD detection using angular distance metric** To test the OOD detection capability of the method proposed in section 3.3, ImageNet-O Hendrycks et al. (2021), SUN Xiao et al. (2010), and aYahoo Farhadi et al. (2009) datasets are used as the OOD datasets. To perform the detection experiment, 2000 data points randomly selected from the ImageNet dataset were mixed with the 6000 data points from ImageNet-O, SUN, and aYahoo datasets (2000 data points from each dataset) and the ID score $S_{ID}$ was computed. We have also compared our method with state-of-the-art contrastive model OOD detection algorithms – KNN+ Sun et al. (2022a) and SSD+ Sehwag et al. (2021). The SSD is a fundamentally different algorithm that predicts few-shot OOD detection. And CSI Tack et al. (2020) is computationally inefficient to implement on a large dataset like ImageNet. In section A.2, figure 9 and table 3 show the distribution and the mean scores for in-distribution and out-of-distribution data points for the proposed exemplar-based method, KNN+, and SSD+.

We have also performed AUROC and FPR@95TPR tests by setting a threshold in the ID score values. The threshold for the proposed method optimal $S_{th}$ is found to be $0.4$. For KNN+, the $k$ hyperparameter and the threshold are set to be $1000$ (for ImageNet) and $-0.72$, respectively, as per the recommendation of including $95\%$ of in-distribution data samples. We used the default parameters for the experiments with SSD+ and a threshold of $700$ is set by empirical analysis. Table 2 shows the FPR@95TPR, AUROC, and ID accuracy scores for the ImageNet-O, SUN, and aYahoo datasets based on the three different methods. Except for the AUROC of the SUN dataset, we can see that our method outperforms in every category.

Table 2: AUROC and FPR@95TPR scores for ImageNet-O, SUN, and aYahoo datasets with KNN+, SSD+, and proposed exemplar-based OOD detection algorithms. In this experiment, ImageNet is considered as the ID dataset.

| | ImageNet-O | | SUN | | aYahoo | | ID Mean |
|---|---|---|---|---|---|---|---|
| | AUROC (↑) | FPR@95 (↓) | AUROC (↑) | FPR@95 (↓) | AUROC (↑) | FPR@95 (↓) | Accuracy |
| Proposed | **70.07** | **85.50** | 72.82 | **80.05** | **68.38** | **86.39** | **64.62** |
| KNN+ | 68.09 | 93.85 | **73.03** | 90.80 | 67.24 | 88.20 | 52.08 |
| SSD+ | 61.63 | 93.89 | 67.41 | 97.10 | 60.91 | 94.09 | 57.88 |

## 4.4 A Sample Report

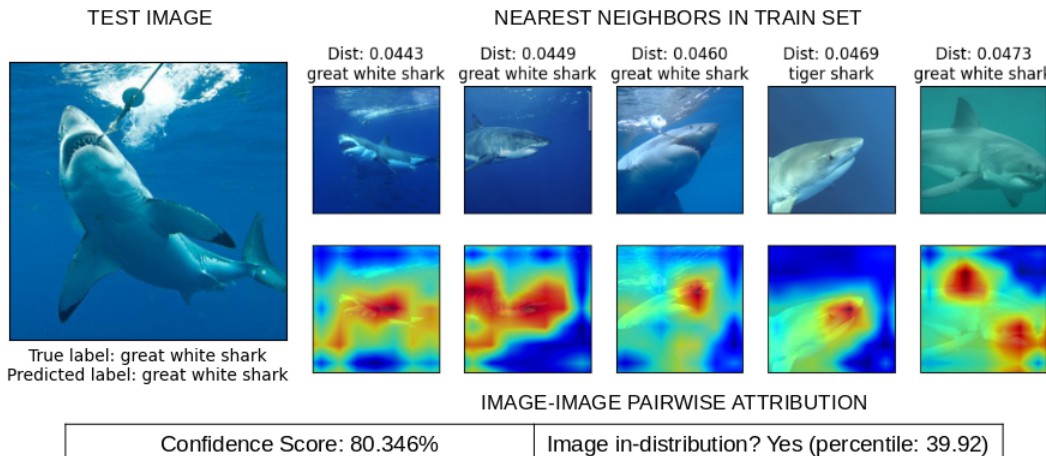

Figure 6: A sample report showing the given test image, image exemplars in the training set and image-image pairwise attribution maps, and confidence and OOD reports showing the result with the proposed framework.

The report from our proposed framework depicted in figure 6 shows that the attribution maps of the exemplars that could identify the correct object in the test image (the most salient features are the shark fin and jaw), the confidence of the label prediction is 80.346%, and the image is in the distribution of the ImageNet dataset (reported ID score $S_{ID}$ is 0.601 and percentile $p_i$ is 39.92 ). The combination of these metrics provide a stronger explanation and trust in the contrastive AI model. A model may sometimes give a very confident prediction outside the model's knowledge limits. The framework ensures that a potential user would be aware of such a situation.

## 5 DISCUSSION

Inspired by the recent success of training visual language models using unsupervised contrastive learning, we proposed new explainable AI techniques based on exemplars. We made the following contributions: (i) explained exemplars using pairwise image-image attributions, (ii) used exemplars to compute confidence scores, and (iii) analyzed knowledge limits using exemplars. The experimental evaluation presented demonstrates the ability of the proposed techniques to provide meaningful explanations for contrastive models. In particular, the attributions generated point out key features between input and exemplars that allow for easier human interpretation of model decision-making. Additionally, the confidence and knowledge limit measures are insightful and provide very strong results with minimal use of empirical parameters. The proposed techniques in this paper are well-positioned to explain existing and future foundational visual language models.

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

# A  APPENDIX

In this appendix, we provide supplementary information and ablation studies that did not fit within the regular paper. The reliability diagrams for SUN and aYahoo datasets are given in section A.1. The distribution of ID scores for OOD detection from different methods is shown in section A.2. We evaluate image attributions with respect to test images in section A.3 and text prompts in section A.4. Additional details of the ODD datasets are provided in Section A.5. Some complete reports are shown for image samples from OOD datasets in section A.6. The environmental impact is analyzed in Section A.7. Lastly, limitations of the proposed exemplar-based explanation method are summarized in Section A.8.

## A.1  RELIABILITY DIAGRAMS FOR SUN AND AYAHOO DATASETS

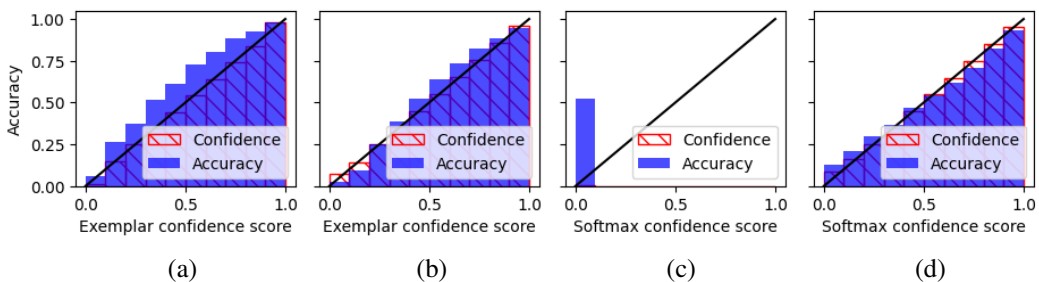

(a)        (b)        (c)        (d)

Figure 7: Reliability diagrams for CLIP image encoder model with SUN dataset: (a) exemplar-based uncalibrated, (b) exemplar-based calibrated, (c) softmax-based uncalibrated, and (d) softmax-based calibrated scores.

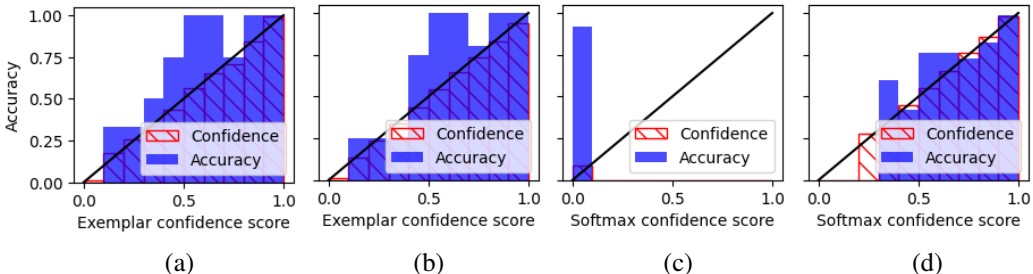

(a)        (b)        (c)        (d)

Figure 8: Reliability diagrams for CLIP image encoder model with aYahoo dataset: (a) exemplar-based uncalibrated, (b) exemplar-based calibrated, (c) softmax-based uncalibrated, and (d) softmax-based calibrated scores.

## A.2  DATA DISTRIBUTIONS FOR OOD DETECTION FOR DIFFERENT DATASETS AND METHODS

Table 3: OOD detection score for in-distribution and out-of-distribution data for KNN+, SSD+, and proposed methods.

|  | **In-distribution** | **Out-of-distribution** | | |
|---|---|---|---|---|
|  | **ImageNet** | **ImageNet-O** | **SUN** | **aYahoo** |
| **Proposed** | 0.46 | 0.25 | 0.22 | 0.27 |
| **KNN+** | $-0.57$ | $-0.62$ | $-0.64$ | $-0.63$ |
| **SSD+** | 803.64 | 913.72 | 905.61 | 929.67 |

## A.3  IMAGE ATTRIBUTION WITH IMAGE PROMPTS

In this section, we show additional examples of exemplar attributions computed with respect to test images. The process of calculating attribution maps is shown in section 3.1. Figures 10 to 13 show

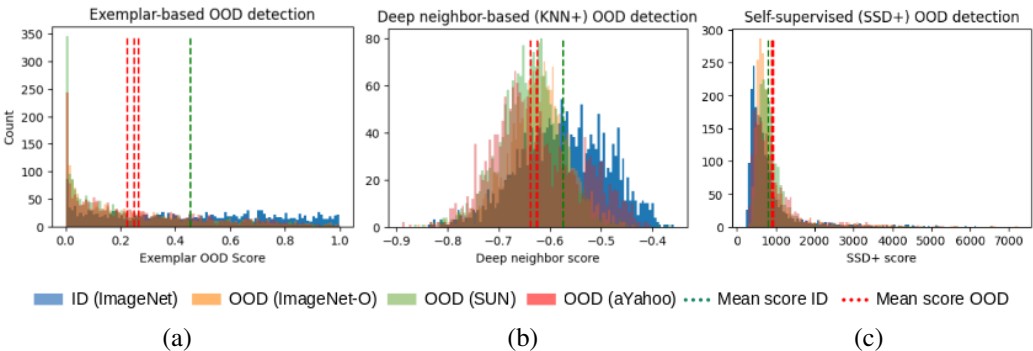

Figure 9: Histogram for the data prediction percentile of ID and OOD data points for (a) Exemplar-based, (b) Deep neighbor KNN+, and (c) Self-supervised SSD+ methods.

the nearest exemplar neighbors based on a given test image prompt and the calculated attribution maps for explaining those exemplars.

Figures 10 and 11 illustrate test images that are properly learned by the model in the experiment (CLIP). The Scorpion in figure 10 the exemplar images has higher saliency around Scorpion regions. One exemplar image of the Garden Spider has higher saliency around the legs of the Spider, which looks visually similar to the scorpion's legs. The Partridge exemplar image in figure 11 also looks visually similar to the Quail in the test image.

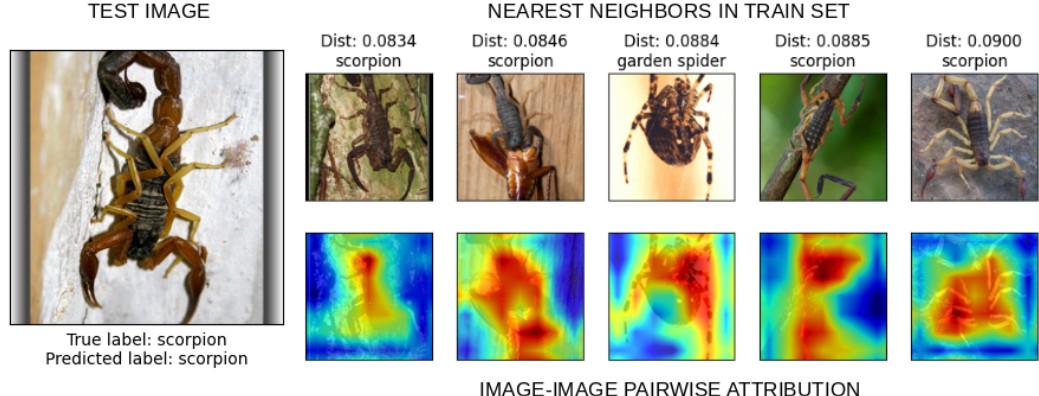

Figure 10: Image-image pairwise attribution with exemplar neighbors.

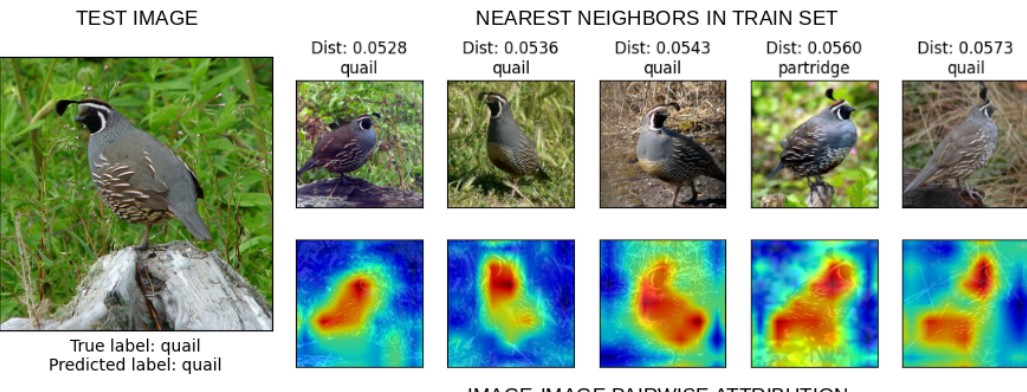

Figure 11: Image-image pairwise attribution with exemplar neighbors.

We also found some examples which are not properly learned by the CLIP model. From the attribution maps in figure 12 we can see that the House Finch label is correctly predicted but the most salient regions are identified to be the sky and the wires. Similarly, in figure 13, the Jellyfish has higher saliency in background seawater.

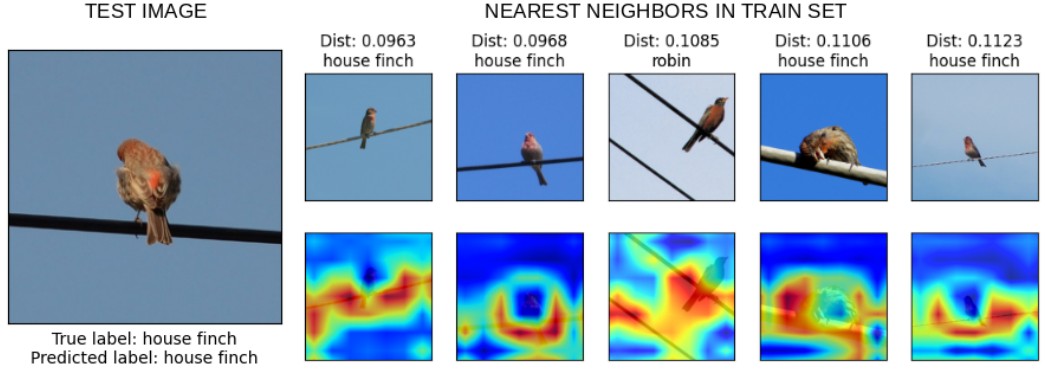

Figure 12: Image-image pairwise attribution with exemplar neighbors.

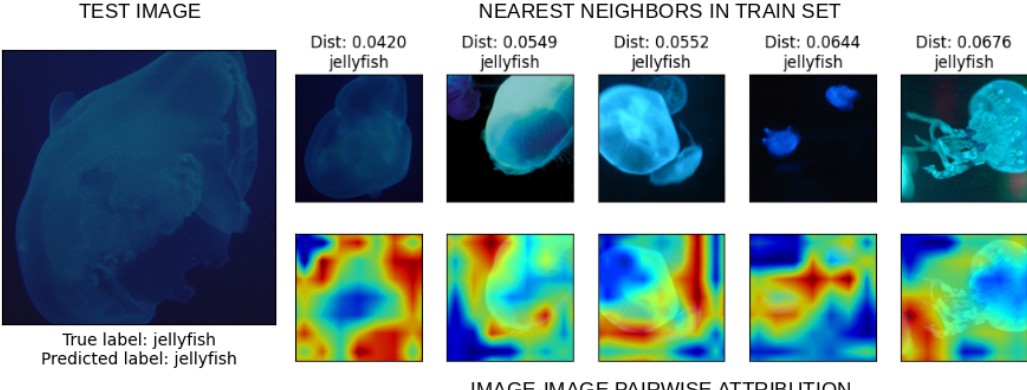

Figure 13: Image-image pairwise attribution with exemplar neighbors.

### A.4 IMAGE ATTRIBUTION WITH TEXT PROMPTS

In this section, we show additional examples of exemplar attributions computed with respect to text prompts. To generate the exemplars of the given text prompt and the attribution of the text data to the exemplar images, a similar approach is used as shown in section 3.1. The process is illustrated in figure 14.

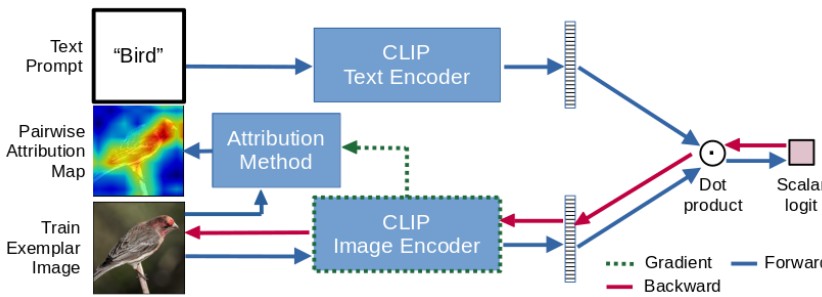

Figure 14: Text-image pairwise attribution method.

Figures 15 to 18 show the nearest exemplar neighbors based on a given text prompt and the calculated attribution maps for explaining those exemplars. Giving text prompts of objects (figures 15 and 16) can properly search the nearest exemplars having the object and/or texts in the image.

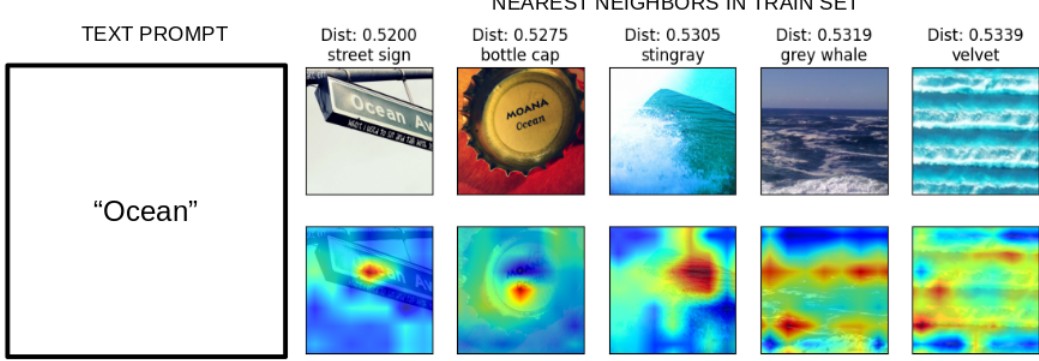

Figure 15: Text-image pairwise attribution with exemplar neighbors.

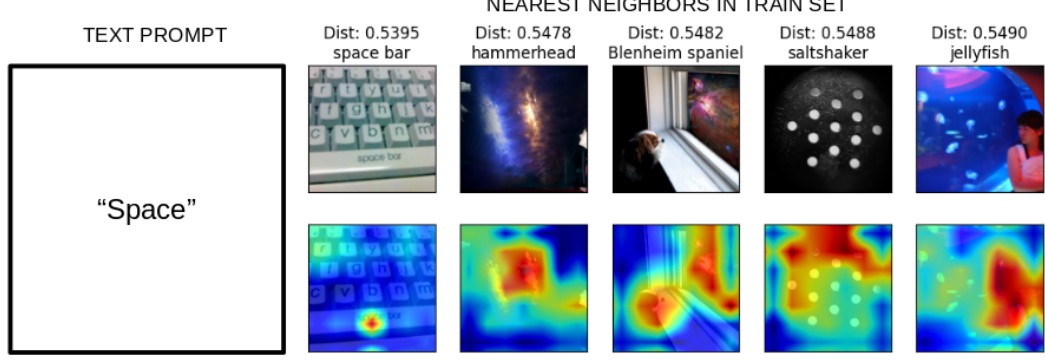

Figure 16: Text-image pairwise attribution with exemplar neighbors.

We have also tested our attribution method with words of abstract ideas (figures 17 and 18). We have seen that the attribution method could identify objects related to the abstract word and could also identify texts in the image corresponding to the given prompt.

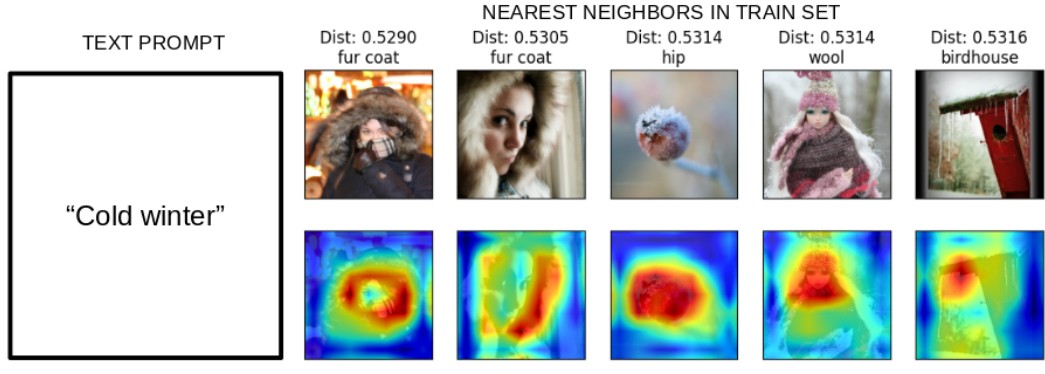

Figure 17: Text-image pairwise attribution with exemplar neighbors.

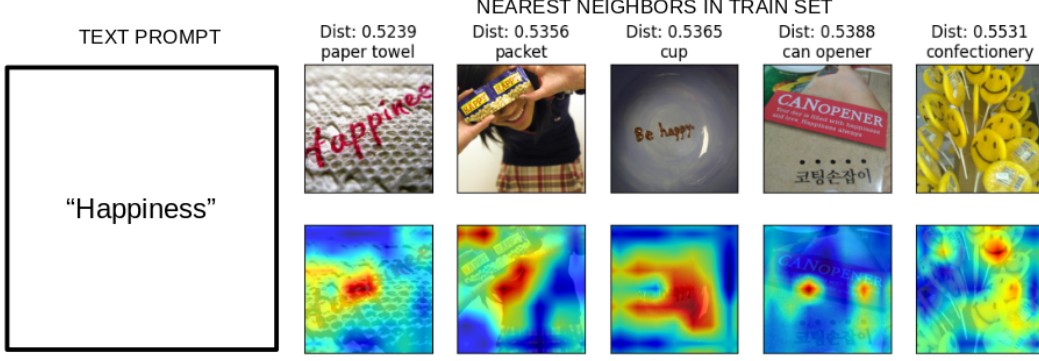

Figure 18: Text-image pairwise attribution with exemplar neighbors.

## A.5 DATASETS USED FOR OOD DETECTION

We used three datasets for OOD analysis – ImageNet-O, SUN, and aYahoo. The SUN and aYahoo datasets are independent datasets for different classification objectives. However, SUN and aYahoo datasets can be considered as out-of-distribution of the ImageNet dataset. On ther other hand, the ImageNet-O dataset is specifically designed for OOD analysis with the ImageNet dataset.

The ImageNet-O dataset is designed with a 200 class subset of 1000 classes from the ImageNet 1K dataset. The selected 200 classes in this adversarially created dataset cover the most broad categories of ImageNet 1K. To create this dataset, images from ImageNet 22k were analyzed. After removing the images of ImageNet 1K from the broad dataset, the images that have a high confidence score from a ResNet50 model-based prediction were kept. Then a subset of high-quality images was selected from the set of images with higher confidence. In total, this dataset contains 2,000 data samples.

## A.6 SAMPLE REPORTS FOR OOD DATA

In this section, we have included sample reports from the three OOD datasets – ImageNet-O, SUN, and aYahoo (figures 19 to 21).

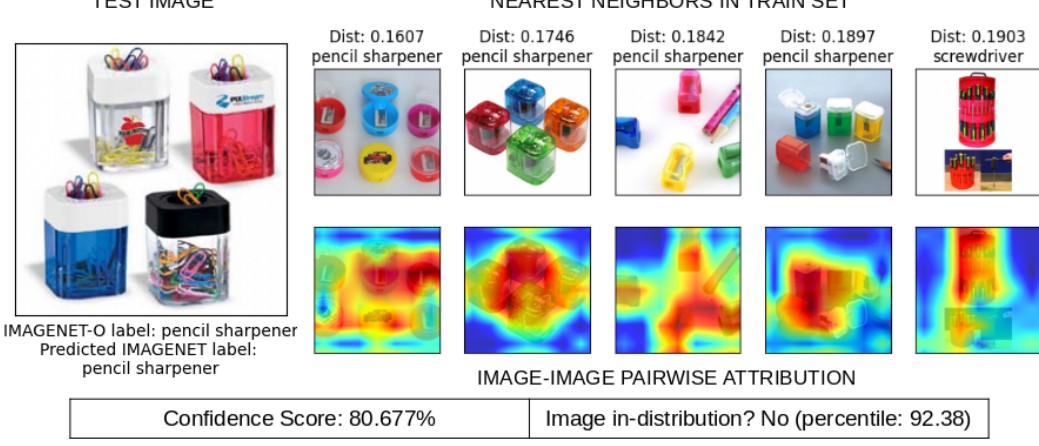

Figure 19: A sample report showing the given test image from ImageNet-O dataset, image exemplars in the ImageNet training set and image-image pairwise attribution maps, and confidence and OOD reports showing the result with the proposed framework.

Here in these experiments, the test image data is obtained from one of the OOD datasets and the exemplars are searched from the ImageNet ID dataset. The confidence and OOD detection are also

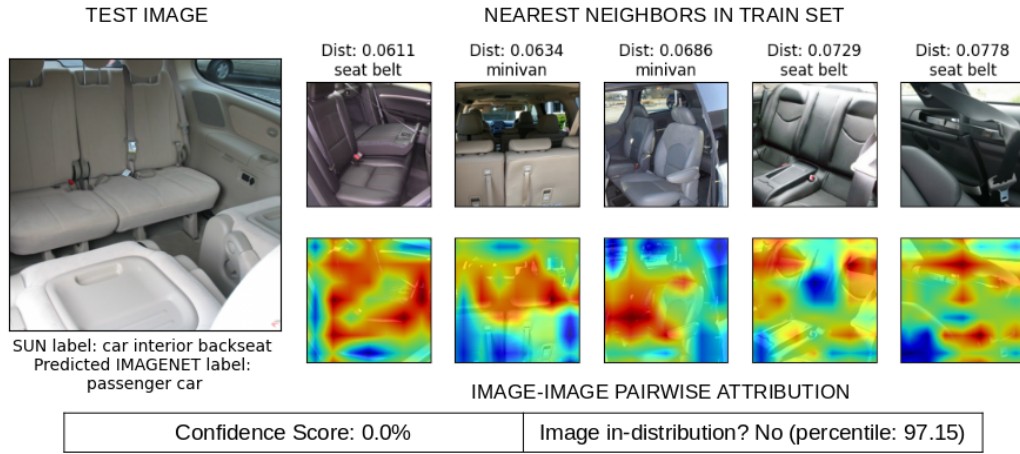

Figure 20: A sample report showing the given test image from SUN dataset, image exemplars in the ImageNet training set and image-image pairwise attribution maps, and confidence and OOD reports showing the result with the proposed framework.

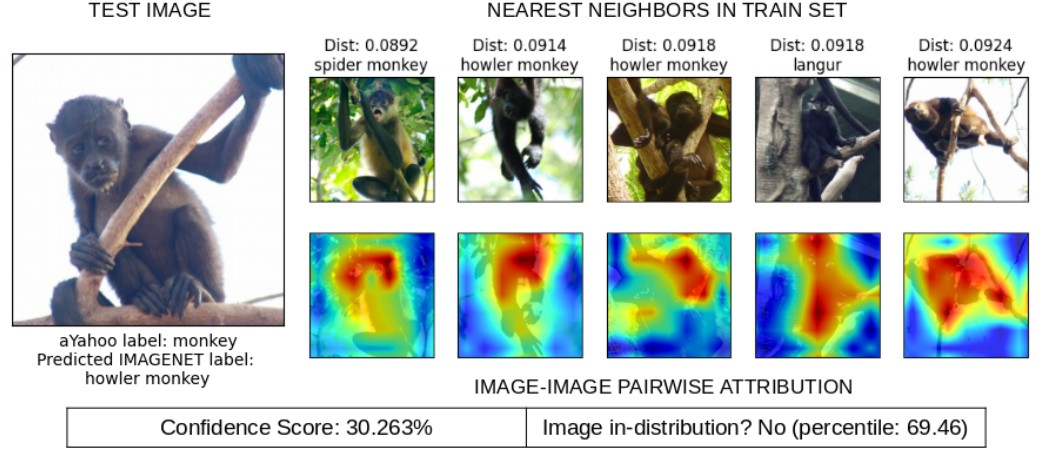

Figure 21: A sample report showing the given test image from aYahoo dataset, image exemplars in the ImageNet training set and image-image pairwise attribution maps, and confidence and OOD reports showing the result with the proposed framework.

computed using the ID exemplars. In figures 19 to 21, we can see that even if the OOD images look visually similar to the ID exemplars, the confidence scores are lower and the images are classified as out-of-distribution of the ID ImageNet dataset.

## A.7 ENVIRONMENTAL IMPACT

Given the computer hardware defined in the manuscript, we provide an approximate calculation of the environmental impact of our experimental evaluation. Given the full evaluation required approximately 6hrs of GPU compute, the evaluation released 0.78kg of $CO_2$ according to ML $CO_2$ impact Lacoste et al. (2019). These numbers are calculated given the default energy efficiency set by ML $CO_2$ impact, and with zero purchased carbon offset.

## A.8 LIMITATIONS

There are several limitations to this method that we think are important to discuss. This proposed method depends on the widely studied exemplar data points. These exemplars are calculated using a k-nearest neighbor algorithm, which is computationally expensive. Though this study does not focus

on the detection of exemplars but rather develops methods based on the exemplars, the calculation of exemplars can impact the overall computational efficiency.

Furthermore, in some cases where the latent space is crowded or has higher density in the latent space with many classes localized into a tiny space, the exemplars extracted for some input may belong to an incorrect class. This results in an inaccurate visualization of the model decisions. We believe this behavior can be explained by two factors. First, imperfections in the data distribution lead to cases of overlap in the dimension of interest. Second, the use of a one-dimensional distance metric means the data that is aligned in the dimension of interest is treated as belonging to the same class.

