# OpenReview forum: "Explaining Contrastive Models using Exemplars: Explanation, Confidence, and Knowledge Limits"
_ICLR.cc/2024/Conference — ICLR 2024 Conference Withdrawn Submission_

### Official Review · Reviewer_VpfW · 2023-10-29

**Soundness:** 3 good
**Presentation:** 2 fair
**Contribution:** 2 fair
**Rating:** 3
**Confidence:** 4

**Summary:**

This paper is trying to address the challenge of explaining "black-box" AI models, emphasizing contrastive learning, which aligns similar data samples in latent space. It introduces:
1. Pairwise attribution analysis for explaining test-training data relationships.
2. Exemplar-based confidence metrics,
3. Measures for model knowledge limits.

**Strengths:**

- The paper introduces confidence metrics and measures for model knowledge limits, making it a complete approach to understand and trust AI models better.
- The improvement in OOD detection is noteworthy. OOD detection is important for real-world applications where the model might encounter unseen or rare data.

**Weaknesses:**

- The "Related works" section should be better organized. The author should summarize works that are directly related to the article and provide concise summaries highlighting points of relevance to this paper.
- Exemplar Selection in Section 3.1: How does the method ensure that the selected exemplars are indeed representative? If the process requires calculating the embedding of all training images first, it could lead to extensive computational overhead.
- The introduction to "AttrMethod" is not clear. Since it's mentioned as "any Attr Method," a preliminary explanation should be provided to the reader for better understanding.
- The entire approach seems to be a combination of various methods (A+B+C). It would be beneficial if the authors could convincingly argue why such a combination is necessary and what unique advantages it brings.
- The explanation presented in section 4.1 is somehow not complete. Instead of providing a support explanation for the current test image, the authors emphasize the attribution map of the exemplars. While this might offer some support, it doesn't adequately explain the decision basis for the current test instance.

**Questions:**

- How does the proposed exemplar-based explanation method compare with other state-of-the-art XAI techniques in terms of accuracy, interpretability, and computational efficiency?
- If OOD instances arise during the pairwise stage, how does the method handle them?
- Why was the CLIP model specifically chosen for this study? What characteristics or features of CLIP made it suitable for this research?
- Why was only the CLIP model used for evaluation? Are there plans to test the proposed method on other models? If not, what limitations or challenges prevent the exploration of other models?

---

### Official Review · Reviewer_CGuX · 2023-10-29

**Soundness:** 3 good
**Presentation:** 3 good
**Contribution:** 3 good
**Rating:** 8
**Confidence:** 3

**Summary:**

This paper proposes a novel exemplars-based XAI method for contrastive learning models, which includes explanations using pairwise attributions analysis, confidence measure using KNN, and in-distribution scoring for knowledge limit explanantion. The experiments demonstrate some interesting explanations and the comparison with the SOTA confirm several advantages of the proposed XAI method.

**Strengths:**

1. The proposed three modules for explaining contrastive learning models are novel and providing new perspectives and methods for interpreting contrastive learning models.

2. The paper is technically sound and provides a detailed and well-structured description of the proposed methods, including the mathematics and algorithms involved. The experimental evaluation, conducted on ImageNet using the OpenAI CLIP model, demonstrates the effectiveness of the proposed techniques.

3. This paper is overall well-written and organized. It clearly explains the concepts, methods, and experimental procedures.

**Weaknesses:**

1. Lack of Comparison with Other XAI Methods: The paper does not compare its proposed techniques with existing XAI methods for contrastive models. A comparative analysis with other approaches would help establish the uniqueness and superiority of the proposed methods.

2. Clarity on Hyperparameters and Thresholds: The paper mentions optimal values for some hyperparameters, such as α and β, but does not provide detailed information on how these values were determined. Providing insights into the hyperparameter tuning process and sensitivity analysis would help readers implement the framework effectively. Moreover, the rationale behind selecting specific threshold values for OOD detection should be explained.

3. Reproducibility and Open Source Implementation: To enhance the reproducibility and adoption of the proposed methods, consider providing open-source code, pre-trained models, and guidelines for researchers and practitioners to apply your framework to their own tasks. This would make it easier for the community to validate and build upon your work.

**Questions:**

1. Can you provide a more comprehensive comparison of your exemplar-based techniques with other XAI methods for contrastive models?

2. Why the attribution analysis is only used for train exemplar image, but that of test image is not given?

3. Can you explain the process by which you determined the optimal values for hyperparameters (e.g., α and β) and thresholds for OOD detection? Is there any sensitivity analysis conducted to show how changes in these values affect the results?

4.  How to combine the explanations from three modules, i.e. attribution analysis, confidence metric, knowledge limit, to form a unified explanation of proposed method ?

**Details Of Ethics Concerns:**

No ethics concerns.

---

### Official Review · Reviewer_NdQo · 2023-10-31

**Soundness:** 3 good
**Presentation:** 3 good
**Contribution:** 2 fair
**Rating:** 5
**Confidence:** 4

**Summary:**

The work leverages **exemplars** within the domain of contrastive vision encoders (specifically CLIP image encoders) towards improving explainability. The work performs pairwise input attribution with NN exemplars, highlighting which features within the exemplar images are relevant to the image under examination. The work proposes a measure of confidence using the distance from exemplars, and the distance from class centroids to characterize OOD samples.

**Strengths:**

- The direction of using exemplars for contrastive models has potential towards improving explainability
- The idea of attributing similarities between images is interesting. However, I wish there had been more discussion and experiments on this part.
- Using distance in using an angular distance metric for OOD detection within the context discussed in this work is interesting. It is shown to be a potent method for this purpose.

**Weaknesses:**

1. The proposed contribution from the perspective of explainability is minimal. It does not reveal new insights. Exemplars is not a new idea; why would it be a contribution to extend it to contrastive models?

2. In principle, Fig.1 or Eq2 is just an attribution of similar features between two images (nearest neighbor images). It does not seem to be relevant to the output of any downstream task. The CLIP encoding of one specific ostrich image is a unique vector, and it’s always (most probably) close to other ostrich images. But this type of analysis, finding the nearest neighbor without considering the target class in any downstream task, always gives the Ostritch image in any task. Consider one of the Ostrich images. Classes, such as trees and grass, are also in the image. Both grass and trees get high clip similarity scores (to text encoder). The method, in that case, better show top neighbors considering these other concepts and attribute to those. The current formulation is limited.

3. It is assumed that ANY attribution approach will work as expected. Unfortunately, this is not the case, as attributions are known to be disagreeing. Each one reveals different information. Thus, it’s best to avoid generalization to ANY attribution method and just state what is done in the paper (which is activation values). It is better to put more detail on how this is exactly computed.

4. More evidence is needed to prove if the attribution is working as expected. It might be that we are just highlighting salient features of the image *irrespective* of the other image. Try using the same Train Exemplar image with test images of other classes. This should be shown systematically, leaving no questions. (though I believe the authors if they let me know they tried and the method is sensitive)

### Minor
- The visual attribution does not give much information. Most of the information regarding explainability comes from exemplars themself which is a previous work.  It would be more useful if it’s more fine-grained with respect to concepts in the images (using CLIP text encoder).
- There are only a few qualitative examples of attribution. The work requires quantitative evaluations (for example, using segmentation masks) or some systematic qualitative experiments to convince us that it is working.
- The confidence score is not a very interpretable number (imagine a user using your approach, use a short description useful to the user)

**Questions:**

- Please focus on addressing the major weaknesses above.
- What makes the pairwise attribution unique to contrastive methods? CLIP encoder is trained contrastively by text-image pairs, not image-image pairs. In short, doing the same similarity attribution should be possible on any image encoder.
- The OOD, with respect to adversarial examples, can be ineffective. You can design perturbed inputs (e.g., ostrich and sharks) that are all close to the embedding space while. Some comments on this would be helpful.

---

### Official Review · Reviewer_fEiN · 2023-11-01

**Soundness:** 2 fair
**Presentation:** 3 good
**Contribution:** 2 fair
**Rating:** 3
**Confidence:** 4

**Summary:**

This paper focuses on enhancing the explainability of AI for contrastive models. The three key contributions are: firstly, it explains the relationship between test and training data using pairwise attribution analysis. Secondly, it introduces exemplar-based confidence metrics. Lastly, it defines measures to determine the extent of a model's knowledge. Furthermore, the proposed exemplar-based confidence score proves the scores are more reliable than traditional methods like softmax score and temperature scaling. Additionally, the paper claims that their proposed framework is eligible to perform the out-of-distribution (OOD) detection. The effectiveness of the framework is demonstrated through extensive experiments.

**Strengths:**

- The paper is well-written and easy to read. The idea and method proposed in this paper are clearly illustrated and introduced, making the reader easily understand.
- The paper discusses an essential question on example selection in contrastive learning via XAI approaches.

**Weaknesses:**

- Emphasizing the salient regions doesn't necessarily lead to clearer explanations. It's not guaranteed that the subject is crucial for prediction. Specifically, an article titled "Sanity Check for Saliency Maps" by (Adebayo et al.) highlights the potential pitfalls of methods like Smoothgrad, as examined in their study. A consistency of salient regions between a pair of exemplars cannot reveal its similarity.
- The authors claim that the proposed framework is adaptive to any attribution methods. However, they only verify the proposed method on GradCAM, which is not sufficient. The authors are highly encouraged to add more experiments under the usage of other attribution methods, such as Smoothgrad and Integrated Gradient.
- In specific contrastive learning frameworks, like He et. al [1], positive examples are chosen by matching input labels. This method is somewhat at odds with measurements based on the concept of saliency maps. For instance, some dog images may display only the dog's head, while others may show the dog's body. In such cases, the saliency maps would vary significantly in their attributions, yet, these images are still chosen as positive examples for the input.
- The framework claims to be well-adaptive in explaining contrastive learning models; however, the proposed framework is only evaluated on the RN101-based CLIP model. The authors are highly encouraged to test their framework on more contrastive learning models, such as a representative one from He et. al [1].

[1] He et. al "Momentum Contrast for Unsupervised Visual Representation Learning" 2020

**Questions:**

- The paper choose to use GradCAM as the explainer to generate the saliency maps. What is the reason of choosing this specific XAI algorithms instead of others that are more powerful and faithful, such as Deepshap [2] and FastSHAP [3].

[2] Lundberg et. al "A unified approach to interpreting model predictions." 2017

[3] Jethani et. al "FastSHAP: Real-Time Shapley Value Estimation" 2021